# Facilitation of Bone Healing Processes Based on the Developmental Function of *Meox2* in Tooth Loss Lesion

**DOI:** 10.3390/ijms21228701

**Published:** 2020-11-18

**Authors:** Tae-Young Kim, Jae-Kyung Park, Yam Prasad Aryal, Eui-Seon Lee, Sanjiv Neupane, Shijin Sung, Elina Pokharel, Chang-Yeol Yeon, Ji-Youn Kim, Jae-Kwang Jung, Hitoshi Yamamoto, Chang-Hyeon An, Youngkyun Lee, Wern-Joo Sohn, Il-Ho Jang, Seo-Young An, Jae-Young Kim

**Affiliations:** 1Department of Biochemistry, School of Dentistry, IHBR, Kyungpook National University, Daegu 41940, Korea; tae09290@gmail.com (T.-Y.K.); yamaryal@yahoo.com (Y.P.A.); euiseon3488@gmail.com (E.-S.L.); shijin1432@gmail.com (S.S.); elinapokharel1996@gmail.com (E.P.); yhs2669@naver.com (C.-Y.Y.); ylee@knu.ac.kr (Y.L.); 2Department of Oral Biochemistry and Molecular Biology, Insititute of Translationnal Dental Sciences, Pusan National University School of Dentistry, Yangsan 50612, Korea; dbgk15@naver.com (J.-K.P.); ilho.jang@pusan.ac.kr (I.-H.J.); 3Department of Biochemistry and Cell Biology, Stony Brook University, Stony Brook, NY 11794-5215, USA; sanjiv.knu@gmail.com; 4Department of Dental Hygiene, College of Health Science, Gachon University, Incheon 21936, Korea; hoho6434@gachon.ac.kr; 5Department of Oral Medicine, School of Dentistry, IHBR, Kyungpook National University, Daegu 41940, Korea; widenmy@knu.ac.kr; 6Department of Histology and Developmental Biology, Tokyo Dental College, Tokyo 101-0061, Japan; hyamamoto@tdc.ac.jp; 7Department of Oral and Maxillofacial Radiology, School of Dentistry, IHBR, Kyungpook National University, Daegu 41940, Korea; chan@knu.ac.kr; 8Pre-Major of Cosmetics and Pharmaceutics, Daegu Haany University, Gyeongsan 38610, Korea; wjsohn@dhu.ac.kr

**Keywords:** periodontitis, signaling pathway, alveolar bone, bone formation, gene therapy

## Abstract

In the present study, we examined the bone healing capacity of *Meox2*, a homeobox gene that plays essential roles in the differentiation of a range of developing tissues, and identified its putative function in palatogenesis. We applied the knocking down of *Meox2* in human periodontal ligament fibroblasts to examine the osteogenic potential of *Meox2*. Additionally, we applied in vivo periodontitis induced experiment to reveal the possible application of *Meox2* knockdown for 1 and 2 weeks in bone healing processes. We examined the detailed histomorphological changes using Masson’s trichrome staining and micro-computed tomography evaluation. Moreover, we observed the localization patterns of various signaling molecules, including α-SMA, CK14, IL-1β, and MPO to examine the altered bone healing processes. Furthermore, we investigated the process of bone formation using immunohistochemistry of Osteocalcin and Runx2. On the basis of the results, we suggest that the knocking down of *Meox2* via the activation of osteoblast and modulation of inflammation would be a plausible answer for bone regeneration as a gene therapy. Additionally, we propose that the purpose-dependent selection and application of developmental regulation genes are important for the functional regeneration of specific tissues and organs, where the pathological condition of tooth loss lesion would be.

## 1. Introduction

The morphogenesis of tooth and supporting tissues is regulated by an intricate network of cell–cell signaling during all steps of tooth development, including the placode, bud, cap, bell, and secretory stages [1,2,3,4,5]. The development processes of tooth and supporting tissues have been highly studied in rodents and revealed the molecular details of signaling interactions between epithelium and mesenchyme [4,5,6,7,8,9]. Particularly, a tooth supporting tissue, periodontium, is originated from dental follicle and to be differentiated into three distinctive structures in the late stage of tooth development, namely, cementum, periodontal ligament, and alveolar bone through complicated signaling networks [10,11,12]. However, these developmental mechanisms involved in the periodontium differentiation are not properly elucidated and remain insufficient for tissue regeneration in clinical approaches [13]. Additionally, most of the studies focused on the examination of signaling molecules and their regulations only in the differentiation of periodontal ligament and cementum, but paid less attention to alveolar bone formation and its regeneration [14,15]. Consequently, problems for the prevention, diagnosis, treatment, and functional regeneration against the disease and traumatic condition of tooth and its supporting tissues, especially in the alveolar bone, still remain unsolved [16,17].

Periodontitis is one of the most prevalent periodontal diseases in an irreversible inflammatory reaction that occurs in the periodontal tissues that support the tooth [18,19,20]. The inflammatory reaction often initiates with the colonization of the bacterial flora in the subgingival surface, which triggers the immune system of host tissues that release cytokines [21,22,23,24,25]. If the spreading inflammation is not prevented, it will worsen the condition by altering the characterization of non-keratinized oral mucosa, hence, degrading the tooth-supporting structures, resulting in the disintegration of the alveolar bone, which, as a severe consequence, leads to tooth loss [1,25]. After the tooth loss due to periodontitis, bone healing capacity is especially necessary for further treatments such as the transplantation of dental implant in clinic [26].

Reciprocal interactions between epithelium and mesenchyme are crucial for various organogeneses, including tooth, hair, mammary gland, and palate [4,6,27]. The epithelial paracrine signaling pathways define the position and regulate the expression of mesenchymal transcription factors, which consequently modulate the paracrine factors and identify the structure of organs [4,5,28]. Among these organs, the palate has been reported to share similar signaling regulations for the determination of the characteristics of palatal mucosa between anterior keratinized and posterior non-keratinized palate [29,30]. Interestingly, *Meox2* (mesenchyme homeobox 2), a vertebrate homeobox-containing transcription factor and one of the mesenchymal signaling molecules for the determination of the soft palatal epithelium along with the inhibition of keratinization and bone formation, has been reported [29,30,31,32,33].

In the present study, we down-regulated the expression level of *Meox2*, in which its developmental function in palatogenesis is revealed and applied into tooth loss lesion due to severe periodontitis for the accommodation of the keratinization of oral mucosa and promotion of the bone formation. We hypothesized that the purpose-dependent selection and application of developmental signaling pathway would be very crucial for the proper functional regeneration of tissues, such as the application of palatogenesis-related signaling molecule to bone healing processes of tooth loss lesion and not to the regeneration of the entire periodontium structure.

In order to evaluate this hypothesis, we applied a range of experimental tools from in vitro to in vivo models and carefully examined altered cytogenesis and histogenesis with signaling regulations after siRNA treatment against *Meox2* into human periodontal ligament fibroblasts (hPDLFs) and mice tooth root socket of tooth loss lesion.

## 2. Results

### 2.1. In Vitro Assays after Knocking Down of Meox2 on Hpdlfs

We applied RT-PCR, 3-(4,5-dimethylthiazol-2-yl)-2,5-diphenyltetrazolium bromide (MTT) assay, and alizarin red S staining (Figure 1) in order to evaluate the knock down ratio, cytotoxicity, and osteogenic differentiation of si*Meox2* transfection on hPDLFs. For 2 weeks, cells were cultivated and harvested for each experiment with control, osteogenic, siControl, and si*Meox2* (Figure 1a). Thirty cycles of PCR showed approximately 20% inhibition of *Meox2* expression when compared with siControl samples (Figure 1b). Statistically, si*Meox2* also showed lower integrated density band than control (Figure 1b). Cells were harvested, and their cell viability and mineralization level were evaluated after 14 days from the transfection of 10 nM siRNA against *Meox2* for 48 h after the seeding. There were no distinctive changes among non-treated, siControl, and si*Meox2* in MTT assay (Figure 1c). Meanwhile, alizarin red S showed an obvious increase of staining levels of calcification in si*Meox2*-treated samples when compared with control and siControl (Figure 1c,d). These results suggest that the knocking down of *Meox2* would promote the osteogenic differentiation of hPDLFs without alterations in cell viability.

### 2.2. Altered Histogenesis and Protein Localizations after the Knockdown of Meox2

Using in vitro hanging drop cultivation with developing upper first molar with palate tissues at E15 for 1 day, the efficacy of siRNA against mouse *Meox2* was also evaluated (Figure 2a). Over 50% of the down-regulated expression of *Meox2* was examined after the siRNA treatment via RT-qPCR (Figure 2a). We also examined the delivery of siRNA against mouse *Meox2* into tooth loss lesion using green fluorescent protein (GFP) expression (Figure 2b,c). GFP positive fluorescent reactions were examined in the tooth root socket after 2 days from siRNA delivery under the fluorescent microscopy (Figure 2c). After these confirmations, we conducted the in vivo experiment of periodontitis-induced animal model for the examination of *Meox2* down-regulation in tooth loss lesion. Briefly, periodontitis was induced by 5-0 silk ligature on maxilla second molar for 5 days at 8 weeks, and then, molar was extracted for mimicking tooth loss lesion [34]. MTC staining and immunostainings against *Meox2* and CK14 were conducted after 1, 2, and 3 weeks from the siRNA treatment in order to investigate the altered histogenesis including new bone formation and protein localizations in the treated tooth root socket and the remaining alveolar bone process regions (Figure 3). After the knocking down of *Meox2*, MTC staining showed a more increased specific localization pattern of newly synthesized collagen bundles in tooth root socket at 1 and 2 weeks (Figure 3a–d). At 3 weeks, si*Meox2*-treated specimens showed the sound bone formation with continuous bone remodeling pattern with newly synthesized collagen (Figure 3e,f). The tooth loss due to the induced periodontitis with disintegrated the entire region of the alveolar bone process showed the obvious tooth root socket, and the formation of collagen (blue color) was examined to confirm the newly mineralized bone tissue in the carrier control and knocking down of *Meox2* specimens for designated periods (Figure 3a–f). The immunolocalization pattern of *Meox2*, after 1 week from the knocking down, was observed in order to investigate the efficiency of siRNA against *Meox2* (Figure 3g,h). Particularly, carrier control group showed an obvious increased positive cell number against *Meox2* in tooth root socket region when compared with si*Meox2*-treated specimens (Figure 3g,h). The localization pattern of Cytokeratin14 (CK14), a well-known intermediate filament marker for keratinization, was observed in order to evaluate the keratinization level of oral mucosa (Figure 3i,j). After 1 week from the knocking down of *Meox2*, the specimen showed a stronger positive reaction against CK14 in the basal layer of oral mucosa than that in the control (Figure 3i,j). On the basis of this altered histogenesis, we hypothesized that si*Meox2* treatment would facilitate the bone formation of tooth loss lesion and the keratinization of oral mucosa around the lesion after 1 week.

### 2.3. Localization Patterns of Osteoblast Related Factor

Immunostainings against Runx2 and Osteocalcin (OC) were conducted in order to examine the activity of osteoblast during the bone healing processes (Figure 4). Much more increased positive cell number against RUNX2 was observed in the knocking down of *Meox2* compared with that in the carrier control at 1 and 2 week (Figure 4a–d, Appendix A). Particularly, significant increased positive cell number against RUNX2 was observed in the apical region of tooth root socket after the knocking down of *Meox2* (Figure 4b′,b″,d′,d″). We also examined the localization pattern of OC, a marker for matured osteoblast. After 1 week, OC was less in the carrier control-treated specimens compared with that in the knocking down of *Meox2* specimens (Figure 4e,f). After 2 weeks from the siRNA treatment, the stronger positive localization pattern of OC was detected in the knocking down of *Meox2* specimens (Figure 4g,h). The stronger positive reactions against Runx2 and OC in *Meox2*-knocking down specimens would suggest that si*Meox2* treatment would facilitate osteoblast differentiation and enhance the bone formation as were examined in in vitro experiment (Figure 1).

### 2.4. Immunolocalization Patterns of Inflammatory Molecules

We applied MPO (Myeloperoxidase), α-SMA (alpha smooth muscle actin), and IL-1β immunostainings after the knocking down of *Meox2* during bone healing processes in tooth loss lesion in order to define the blood vessel formation and inflammation modulations (Figure 5). The stronger positive reaction against MPO was investigated in si*Meox2*-treated group at 1 week when than in the control (Figure 5a,b). The knocking down of si*Meox2* specimens also showed stronger positive reactions against MPO than in the carrier control after 2 weeks (Figure 5c,d). In order to define the blood vessel formation, α-SMA localization pattern was examined in the tooth loss lesion (Figure 5e–h). The similar positive reaction against α-SMA was examined in both the carrier control and the knocking down of *Meox2* after 1 week (Figure 5e,f). However, at 2 weeks from si*Meox2* treatment showed much increased positive reaction against α-SMA in the knocking down of specimens when compared with the control (Figure 5g,h). Additionally, after 1 and 2 weeks of the knocking down of *Meox2*, the localization of the pro-inflammatory molecule IL-1β showed stronger positive reaction in the carrier control group than that of the knocking down of *Meox2* group (Figure 5i–l). Particularly, the upper side region of tooth root socket showed stronger positive reaction against IL-1β in carrier control-treated specimen than in the knocking down of *Meox2* (Figure 5i′,j′) at 1 week. These specific localization patterns of MPO, α-SMA, and IL-1β would suggest that si*Meox2* treatment rapidly altered the cellular physiology, such as the vascularization and inflammatory cell migration into the tooth root socket for the establishment of better environment for bone formation.

### 2.5. Micro-CT Evaluations of Bone Formation after siRNA Delivery

After 2 weeks of knocking down, the precise bone healing capacity was evaluated, following the induction of periodontitis, using Micro-CT evaluation (Figure 6a,b). The volume of bone formation was estimated by Micro-CT at 2 weeks with modifications based on previous reports [34]. Micro-CT images of the new bone volume formed after carrier control and si*Meox2* treatment were evaluated using CTAn (Figure 6). Compared with carrier control-treated specimens, the knocking down of *Meox2* specimens showed dramatic increased volume of bone formation after 2 weeks (Figure 6a,b). The bone formation volume after the knocking down of *Meox2* showed 71.6%, whereas the carrier control-treated specimen showed 54.5% after 2 weeks as determined by CTAn software (Bruker Micro-CT). The analysis revealed that there is an obvious increase of new bone volume after si*Meox2* treatment, which further supported the view that si*Meox2* treatment enhances the bone forming capacity.

## 3. Discussion

### 3.1. Periodontal Disease and Various Clinical Approaches

Periodontium is the most difficult tissues that are complex and its developmental signaling pathways for proper regeneration remain not elucidated. Periodontitis is a serious disease that infect tooth-supporting tissues, destroying and damaging the soft and hard tissues, which can lead to tooth loss if not properly treated [18,19]. Thus far, periodontitis treatment is classified with non-surgical and surgical approaches [35,36,37]. Among the non-surgical treatments, scaling removes tartar and bacteria from tooth surfaces and beneath the gums and root planning smoothens the byproducts that delay the healing or the reattachment of the gums to the tooth surfaces [38,39,40]. Meanwhile, in advanced periodontitis, treatment may require dental surgery, such as flap surgery [36]. These surgical approaches have risks; therefore, non-surgical approaches are needed for in restoring the bone by applying dental implants, which require formed bone in the exfoliated sockets for which invasive surgical approaches of grafting bones have been practiced for a long time already [41,42]. The only way of non-surgical approach is to develop the novel therapeutic compounds that can be applied in the affected area to stimulate the signaling molecules that induce the stem cells in the residual periosteal or periodontal ligament tissues in order to differentiate into osteogenic cells or direct gene therapy to facilitate the stem cell differentiation into osteogenic fate [43,44]. To date, many studies have been conducted to test the therapeutic compounds for preventing bone loss and regeneration of bony tissue [45,46,47], but very little research has reported about gene therapy [47,48]. For overcoming the periodontitis, the pathological signaling pathways involved in bone healing capacity must be understood and the target molecules for new drug development must be discovered.

### 3.2. Application of Developmental Mechanisms on Functional Restoration

We hypothesized that the inhibition of soft palate formation related gene, *Meox2* (mesenchyme homeobox 2) would cause the osteogenic potential in tooth loss lesion. To elucidate this theoretical approach for hard tissue formation, we applied an in vitro cell culture using hPDLFs. Specifically, alizarin red S staining at 14 days, after the knocking down of *Meox2* for 48 h in early cultivation on hPDLFs, showed dramatic osteogenic differentiation when compared with siControl and osteomedium only conditions (Figure 1d). This osteogenic differentiation potential would be similar to previous reports, which showed dramatic increase of bone formation through Runx2 and OC signaling [39,49]. We also examined the possibilities of increased *Meox2* localization in tooth loss lesion due to the severe periodontitis. Immunohistochemistry results showed obvious increase of *Meox2* positive cells in tooth root socket (Figure 3g,h). This specific and pathologic localization of *Meox2* was not revealed in developmental condition of periodontium tissue and even in knockout mice studies, since tooth loss lesion would be totally different from the normal site, as expected [18,19,30,50]. Previous study, using *Meox2* knockout mouse, only reported reduction of muscle mass and loss of specific muscle types primarily from the limbs [50]. In addition, based on our previous report, we hypothesized that pathological conditions after the tooth loss from the periodontitis would share similarities in morphological aspects with palate developing tissue [29,34]. Soft palate formation region showed only muscle and fiber formation rather than bone formation with a higher expression level of *Meox2* [29,30].

### 3.3. Facilitation of Bone Formation by Knocking Down of Meox2

We investigated the efficacy of the knocking down of *Meox2* after 1 and 2 weeks in tooth root socket following periodontitis for identification of *Meox2* function in bone formation and oral mucosa keratinization, which would be aims for further clinical applications, including dental implants and inflammation controls. For these purposes, we applied the induced periodontitis animal model system, established in the previous report [34], and this experimental system enables us to determine potential candidate gene therapy that can be used for bone forming and hard tissue regeneration time investments for alternative to surgical approach discoveries hence provides a faster route to clinical trials [15]. Then, we carefully examined histomorphology and immunohistochemistry of the signaling molecules related to inflammation and bone formation in tooth root socket (Figure 2, Figure 3, Figure 4, Figure 5 and Figure 6). Because there were copious reports on the relationships between the modulation of inflammation and enhancement of bone formation in tooth loss lesion [21,34,51,52]. The histomorphological examination with MTC staining after knocking down of *Meox2* revealed significant changes (Figure 3a–f). We also observed after 1 and 2 weeks, an increased in the localization of collagen fibers in the tooth root socket and alveolar bone process region of knocking down of *Meox2* (Figure 3a–d). The staining pattern of MTC suggested that knocking down of *Meox2*-induced bone formation in the tooth root socket region after 1, 2, and 3 weeks. Additionally, the knocking down of *Meox2* specimens showed region specific localization of RUNX2, increased in the upper side of tooth root socket after 1 week (Figure 4a′–b″). OC is associated with mature osteoblast [53], and the detection of OC may allow the assessment of bone formation in tooth root socket [34]. After 1 week, the basal side close to the tooth root socket upper side showed positive reaction of OC (Figure 4e′–f′), in which localization was consistent with the MTC staining pattern, indicating increased collagen formation [34]. We also compared localization patterns after 2 weeks; RUNX2 and OC were observed to have similar positive reaction at basal side (Figure 4d′,d″,h′,h″). These specific localization patterns suggest that the new bone formation begins from the existing osteoblastic precursor cells in the tooth root socket, which then undergoes rapid proliferation and differentiation in the presence vascular supply. We also conducted Micro-CT after 2 weeks of treatment in order to examine the precise level of bone formation [34]. The analysis revealed increased new bone volume after si*Meox2* treatment, which further supported the view that si*Meox2* treatment enhances the bon forming capacity (Figure 6). Additionally, we conducted TRAP staining to evaluate quality of bone formation by measuring the number of osteoclasts following si*Meox2* treatment (Appendix A) [54,55,56]. We observed significant increase in osteoclast and Runx2 positive cell number after 1 and 2 weeks of si*Meox2* treatment, which suggests that osteoclast apoptosis induced by osteoblasts (Figure 4, Appendix A) [57]. On the basis of these results, we conclude that *Meox2* knockdown would facilitate the sound bone formation.

### 3.4. Functional Evaluation of Meox2 for Modulating Inflammation

In periodontitis, if the spreading inflammation is not prevented, it will worsen the condition, thereby degrading the tooth supporting structures, resulting in disintegration of the alveolar bone, which, as a severe consequence, leads to tooth loss [25,58,59]. In the case of tooth loss, the quality of life is impaired, and the ultimate option is to have dental implantation [60]. As previously reported, we observed a significant relationship between bone formation and inflammation modulations [34,58,59,61,62]. The process of bone resorption and formation is regulated by the immune system through the secretion of various cytokines, with vascularization playing a significant role [61,62]. The prevention of inflammation would result in the attenuation of bone loss [62]. Furthermore, *Meox2* inhibits in vitro proliferation of vascular smooth muscle and endothelial cells by inducing expression of p21 [63]. At the early stage of si*Meox2* treatment, the obvious increased number of MPO positive cells with more vascular structures in the tooth loss lesion were shown (Figure 3 and Figure 5). As were examined in the previous reports, these fast modulations of inflammation with MPO with blood vessel formation would result into minimizing the IL-1β cytokine localization in later stage of tooth loss lesion (Figure 5) [58,64]. This fast modulated inflammation via neutrophil involvement would contribute the establishment of bone formation circumstances [34,58,59,61,62]. Overall, although we did not examine the detailed modulation mechanisms underlying inflammation control, our histology and immunohistostainings results would suggest that *Meox2* involves in bone formation via inflammation modulations.

### 3.5. Tissue Regeneration Using Developmental Regulating Gene Application

Gene therapy could be one of the best alternatives to induce osteogenic fate in the residual stem cells as all cellular mechanisms are controlled by signaling molecules that activate various pathways [65]. These approaches should be followed and designed by well-established developmental mechanisms during organogenesis for proper regeneration. In these cases, we must consider that simple and direct applications of signaling regulations from the same origin tissue would not be good trials because damaged and destroyed tissues and organs have totally different conditions from the normal developing tissue and organs. For example, the tooth root socket after the tooth loss from the periodontitis is no longer a typical periodontium tissues; instead, this pathological condition is more similar to developing palate, since all the important events for the developing palate are bone formation and/or oral mucosa keratinization. In this study, we have demonstrated the detailed function of *Meox2* and its related signaling pathways involved in bone healing processes using siRNA treatment against *Meox2* in the extraction socket by local delivery method as a gene therapy after tooth loss following periodontitis induction. The developmental regulating gene application for tissue regeneration would be an excellent model for regenerative medicine field. Moreover, the identification of signaling regulations in pathological condition would be essential for the discovery of the target molecules in drug development.

## 4. Materials and Methods

All animal experiments were ethically approved and conducted in accordance with the guidelines of the Intramural Animal Use and Care Committee of Kyungpook National University, School of Dentistry (KNU 2020-0107; 23 July 2020).

### 4.1. In Vitro Cell Cultivation Experiment

#### 4.1.1. Cell Culture and siRNA Transfection

hPDLF cells were purchased from LONZA (LONZA, Walkersvile, MD, USA, cat no. CC-7049) and maintained in SCGMTM stromal cell basal medium (LONZA, Walkersvile, MD, USA, cat no. CC-3204) supplemented with stromal cell growth medium singleQuots^TM^ supplements and growth factors kit (LONZA, Walkersvile, MD, USA, cat no. CC-4181), comprising fetal bovine serum (FBS) (25 mL), insulin (0.5 mL), hFGF-B (0.5 mL), and GA-1000 (0.5 mL), in the humidified incubator with a 5% CO_2_ at 37 °C. *Meox2* siRNAs and control siRNAs were purchased from Origene (Origene Technologies Inc., Rockville, MD, USA, cat no. SR302869). Individual siRNA (20 nM) was transfected into hPDLF cells using Jetprime transfection reagent (Polyplus transfection, USA) in accordance with the manufacturer’s instructions. Briefly, 1.5 × 10^5^ cells/well were seeded in six-well plates and incubated overnight. When the cell density reached 70–80%, siRNA-Jetprime solution and fresh growth medium were gently mixed and added into six-well plates, followed by incubation for 48 h prior to the further experiments.

#### 4.1.2. Osteogenic Differentiation

For differentiation toward osteogenic lineage, 3 × 10^4^ hPDLF cells were seeded into 48-well plates and cultured in the osteoblast differentiation media: α-MEM supplemented with 10% FBS, antibiotic-antimycotic (Biowest, Nuaillé, France, cat no. L0010-100), 50 µM ascorbic acids (Sigma-Aldrich, St. Louis, MO, USA, cat no. A4544), 10 mM β-glycerophosphate (Sigma-Aldrich, St. Louis, MO, USA, cat no. G9422), and 100 nM dexamethasone (Sigma-Aldrich, St. Louis, MO, USA, cat no. D4902). Cells were fixed with 4% paraformaldehyde and stained with alizarin red S (Sigma Aldrich, St. Louis, MO, USA, cat no. TMS-008-C) solution of 2% *w/v* diluted by deionized water at pH 4.3 for 20 min to assess the mineralization level. The stained samples were destained and dissolved in 10% cetylpyridinium chloride solution in 10 mM PBS (pH 7.0) to quantify the calcium deposition. The absorbance at 562 nm was identified using microplate spectrophotometer (Biotek Instruments, Winooski, VT, USA).

#### 4.1.3. Proliferation Assay

hPDLFs viability was determined using 3-(4,5-dimethylthiazol-2-yl)-2,5-diphenyltetrazolium bromide (MTT) assay. The hPDLFs were seeded in 48-well plates (Corning Incorporated, Corning, NY, USA) having a density of 2 × 10^4^ cells/well in a humidified atmosphere with 5% CO_2_ at 37 °C. After the subjection to varying experimental conditions, cells were washed with PBS and incubated with 200 µL of 0.5 mg/mL MTT (Merck KGaA, Darmstadt, Germany) for 2 h at 37 °C. Cells were rinsed with PBS, and the formazan products were dissolved in 200 µL of dimethyl sulfoxide (Sigma-Aldrich, St. Louis, MO, USA) for 15 min at room temperature with agitation. The absorbance at 540 nm was determined using microplate spectrophotometer (Biotek Instruments, Winooski, VT, USA).

### 4.2. In Vivo Animal Experiment

#### 4.2.1. Animals

Adult 8 week-old male B6 mice were used in the experiment (*n* = 24). The mice were assigned as the control group (*n* = 4 per week, a total of 12 for 3 weeks) or the experimental group (*n* = 4 per week, a total of 12 for 3 weeks) with treatment following ligature-induced periodontitis and tooth extraction under anesthesia. Three mice from each of groups were randomly selected and examined the Micro-CT evaluation on the second week first, then processed for further evaluation including histology and immunohistochemistry (Appendix A).The mice were housed by group under the following conditions: 22 °C ± 2 °C, 55% ± 5% humidity, and artificial illumination lit between 05:00 and 17:00 h. Food and water were given ad libitum.

#### 4.2.2. Periodontitis Model System and Tooth Extraction

All mice were ligated with black silk 5-0 (AILEE CO, LTD, Korea, cat no. SK54510) around the maxillary left second molar with a slight modification of the previously described procedure under anesthesia in order to induce periodontitis [51]. The animals were anesthetized via an intraperitoneal injection of Avertin (Sigma-Aldrich, USA), as previously described [66]. After 5 days, the ligature was removed, and the second molar was extracted, which formed the extraction socket [34].

#### 4.2.3. siRNA Delivery in the Extraction Socket

The extraction socket was treated with 2 µL of mixture that comprise Pluronic^®^ F-127 (Sigma-Aldrich, Steinheim, Germany, cat no. CAS: 9003-11-6), 0.05% dimethyl sulfoxide (DMSO; Duchefa Biochemie, Haarlem, The Netherlands, cat no. CAS: 67-68-5), siTran transfection reagent (Origene Technologies Inc. Rockville, MD, USA cat no. TT30001), and scrambled siRNA (negative control) (Origene Technologies Inc. Rockville, MD, USA, cat no. SR30004) for control groups and 2 µL mixture of *Meox2* siRNA (Origene Technologies Inc. Rockville, MD, USA, cat no. SR408862), siTran transfection reagent, Pluronic^®^ F-127, and 0.05% DMSO for the siRNA therapy experimental group. The final concentration of siRNA in the mixture was maintained at 100 nM. To deliver the chemical into the socket, gas-tight Hamilton syringe 84877 (Hamilton, Reno, NV, USA) was used. The area was sealed using fibrin sealant (Tisseel; Baxter, Deerfield, IL, USA, cat no. SKU 1506079) after the delivery of treatment material into the socket in order to ensure that the treatment material is retained in the socket.

#### 4.2.4. Histology and Immunostaining

Histomorphological analyses were conducted using Masson’s trichrome (MTC) staining after fixation and decalcification, as previously described [67]. Immunostaining was conducted, as described previously [34,68]. Primary antibodies against α-SMA (Abcam, Cambridge, UK, 1:100; cat no. ab17534), CK14 (Abcam, Cambridge, UK, 1:100; cat no. ab181519), IL-1β (Abcam, Cambridge, UK, 1:100; cat no. ab9722), MPO (Bioss, Boston, MA, USA, 1:100; cat no. bs-4943R), *Meox2* (Novous biology, Centinnial, CO, USA, 1:100; cat no. NBP2-30647), Osteocalcin (OC; biorbyt, St. Louis, MO, USA, 1:200; cat no. orb309273) and Runx2 (Abcam, Cambridge, UK, 1:1000; cat no. ab192256) were utilized. Secondary antibodies were used as biotinylated anti-rabbit or anti-rat IgG. Binding of the primary antibody to the sections was visualized using the diaminobenzidine tetrahydrochloride reagent kit (GBI Labs, Bothell, WA, USA, cat no. C09-12).

#### 4.2.5. Micro-CT Evaluation

As described previously, micro-computed tomography (Micro-CT) and analysis were conducted [34]. The maxillas were harvested at day 0 before treatment just after the extraction of tooth and 2 weeks after treatment and subjection to Micro-CT. Using a Skyscan-1173 Micro-CT (Bruker MicroCT, Kontich, Belgium), each sample was scanned with the following parameters: 90 kV tube voltage, 88 μA current, 1.0 mm aluminum filter, 9.59 μm image pixel size, 500 ms exposure time, 0.3 rotation step, 360° scan, and four frames per rotation. Using NRecon software (Bruker MicroCT, Kontich, Belgium), three-dimensional (3-D) reconstructions were conducted. Two-dimensional (2-D) and 3-D models of bone tissue formed in the extraction socket area in each sample were created using DataViewer (Bruker MicroCT, Kontich, Belgium) and CTVol software (Bruker MicroCT, Kontich, Belgium), respectively. The percent bone volume (calculated as the bone volume/tissue volume) of the new bone formed in the socket area was determined using CTAn software (Bruker MicroCT, Kontich, Belgium).

### 4.3. In Vitro Mesenchymal Tissue Cultivation and RT-qPCR Evaluation

After the dental mesenchymal tissue collection from developing upper molar at E15, we applied the hanging-drop in vitro cultivation with or without treatment of siRNA for one day, and harvested the in vitro cultivated dental tissues for RT-qPCR analysis, as previously described. Briefly, dental epithelium was removed from the mesenchyme of upper molar tooth germ, following 2.4 units of Dispase II treatment. At least 16 upper molar tooth germs were collected. The collected mesenchymal tissues were each placed in the center of the lid of tissue culture dish with 70 µL of Dulbecco’s modified Eagles Medium (DMEM; HyClone, South Logan, Utah, USA, cat. no. SH30243.01) containing 10% FBS (Gibco^®^, Life Technologies, Grand Island, NY, USA, cat. no. 26140-079) and 1% penicillinstreptomycin (Gibco^®^, Life Technologies, Grand Island, NY, USA, cat. no. 15140-122) with or without 100 nM of *Meox2* siRNA (Origene Technologies Inc. Rockville, MD, USA, cat no. SR408862), using siTran transfection reagent (Origene Technologies Inc. Rockville, MD, USA, cat no. TT30001), and the lid was inverted so that the drop of medium with mesenchymal tissue was left hanging under the lid. The lid with a hanging drop was placed over the culture dish with 1 mL of medium to prevent from drying and incubated for 24 h at 37 °C in the presence of a gas mixture of 50% O_2_, 45% N_2_, and 5% CO_2_. Mesenchymal tissues were harvested, and total RNA was extracted using the RNeasy^®^ Micro Kit (Qiagen, Hilden, Germany, cat no. 74004). Using the Omniscript^®^ RT Kit (Qiagen, Hilden, Germany, cat no. 205111), complimentary DNA was synthesized. Real-time quantitative polymerase chain reaction (RT-qPCR) was conducted using the Step One Plus Real-time PCR system (Applied Biosystems) with the SYBR Green PCR master mix (Applied Biosystems). The results of RT-qPCR for each sample were normalized to hypoxanthine phosphoribosyltransferase (Hprt). The results are expressed as the normalized rations or as mean ± SD. All experiments were repeated at least three times.

### 4.4. Photography and Image Analysis

All slides stained for histology and immunohistochemistry were photographed using a DM2500 microscope (Leica, Wetzlar, Germany) and digital CCD camera (Leica, Wetzlar, Germany, cat no. DFC310 FX). Data are expressed as mean ± SD. Comparisons were made between the experimental and control groups using Student’s *t* test. A *p*-value of <0.05 was considered significant. As described previously, all statistical analyses were conducted using imageJ software [34].

## 5. Conclusions

Based on our results, we suggest that the knocking down of *Meox2* via the activation of osteoblast and modulation of inflammation would be a plausible answer for bone regeneration as a gene therapy. This study provides important principle dependent application of developmental mechanisms into tissue regeneration would be an excellent model system and gene therapy development for overcoming unsolved diseases.

## Figures and Tables

**Figure 1 ijms-21-08701-f001:**
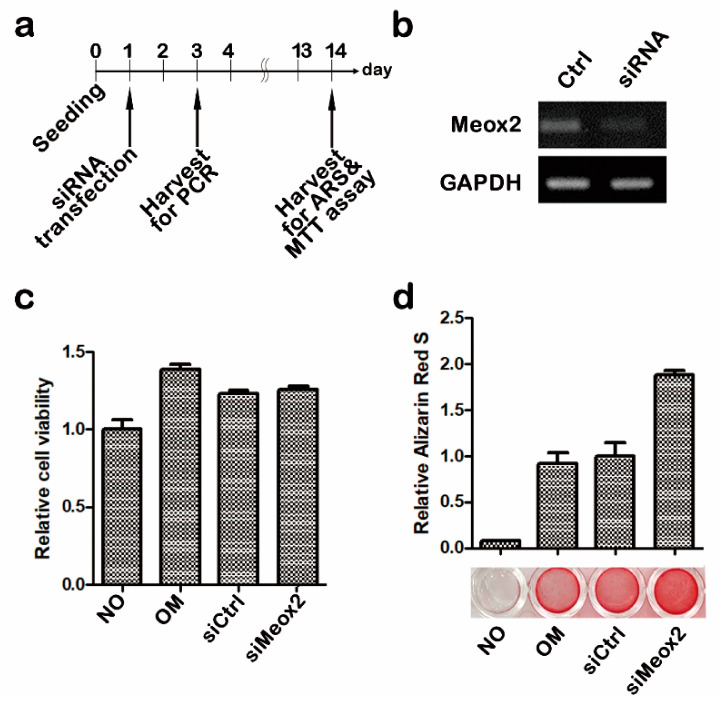
Effect of *Meox2* on the viability of human periodontal ligament fibroblasts (hPDLF) (**a**–**d**). Showing of schematic diagram of in vitro cell cultivation schedule (**a**). The relative expression of si*Meox2* was analyzed using qRT-PCR (**b**). Cell viability of hPDLFs were analyzed MTT assay. hPDLFs were treated with siRNA against *Meox2* (10 nM) for 48 h (**c**). Alizarin red S staining of hPDLFs under 48 h of osteogenic culture with si*Meox2* (**d**). Alizarin red S staining was quantified by cetylpyridinium chloride extraction and measuring the absorbance at 562 nm.

**Figure 2 ijms-21-08701-f002:**
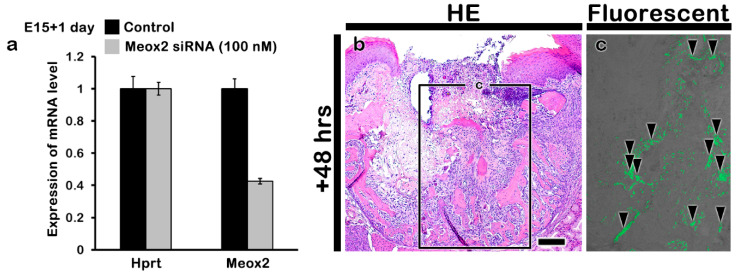
RT-qPCR evaluation of expression pattern of si*Meox2* after 1 day using hanging-drop cultivation with tooth germ together posterior palate at E15 (**a**). Local delivery of siRNA into tooth loss lesion using green fluorescent protein (GFP) expression (**b**,**c**). Black arrowheads in c indicate GFP positive fluorescent reactions.

**Figure 3 ijms-21-08701-f003:**
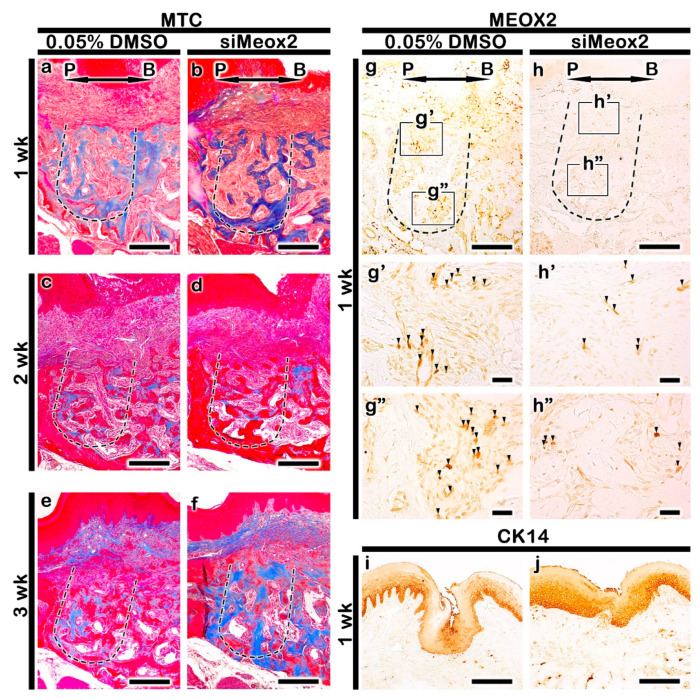
Expression pattern of *Meox2* (**g**,**h**) in tooth root socket, MTC (**a**–**f**) and CK14 (**i**,**j**) staining showing the formation of collagen and keratinization. After 1, 2, and 3 weeks of the knocking down of *Meox2*, the carrier control (DMSO)-treated specimens (**a**,**c**,**e**) have reduced volume of collagen (blue color) compared with that in the knocking down of *Meox2* (**b**,**d**,**f**).The keratinization in carrier control treated specimens is reduced compared with that in si*Meox2* (**i**,**j**). P, palatal and B, buccal. Dotted line demarcates the root region of the socket. The solid boxes depict higher magnification views (**g**′–**h**″). Scale bars 200 µm (**a**–**j**), 100 µm (**g**′,**g**″,**h**′,**h**″). Black arrowheads in g’h’-g”h” indicate Meox2 positive cells.

**Figure 4 ijms-21-08701-f004:**
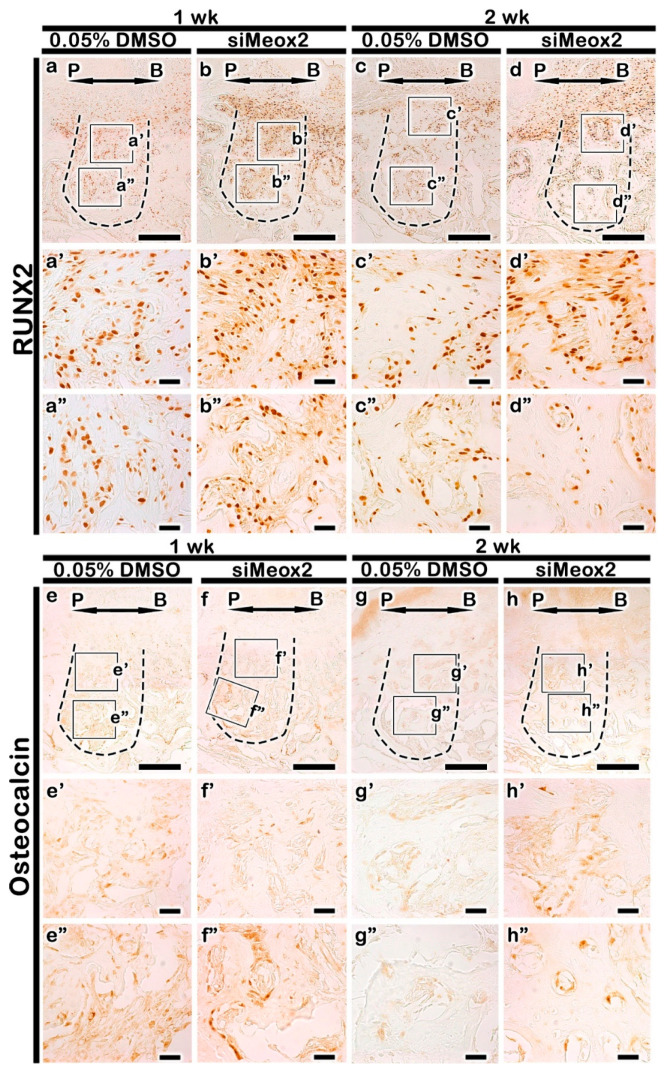
Immunolocalization of pre-osteoblast (**a**–**d**) and mature osteoblast (**e**–**h**) during bone formation. After 1 week of treatment, the carrier control (DMSO)-treated specimens showed a increase in Runx2 level (**a**) compared with that in si*Meox2* specimens (**b**). The number of Runx2 positive cells lower in carrier control-treated specimens (**c**) than in si*Meox2* treated specimens (**d**) after 2 weeks of treatment. After 1 week of treatment, the carrier control (DMSO)- treated specimens show increase in OC level (**e**) compared to si*Meox2* treated specimens (**f**). OC is observe higher positive cells in si*Meox2* treated specimens (**g**) than in carrier control-treated specimens (**h**). The solid boxes depict higher magnification views. P, palatal and B, buccal. Dotted line demarcates the root region of the socket. The solid boxes depict higher magnification views (**g**′–**h**″). Scale bars 200 µm (**a**–**h**), 100 µm (**a**′–**h**″).

**Figure 5 ijms-21-08701-f005:**
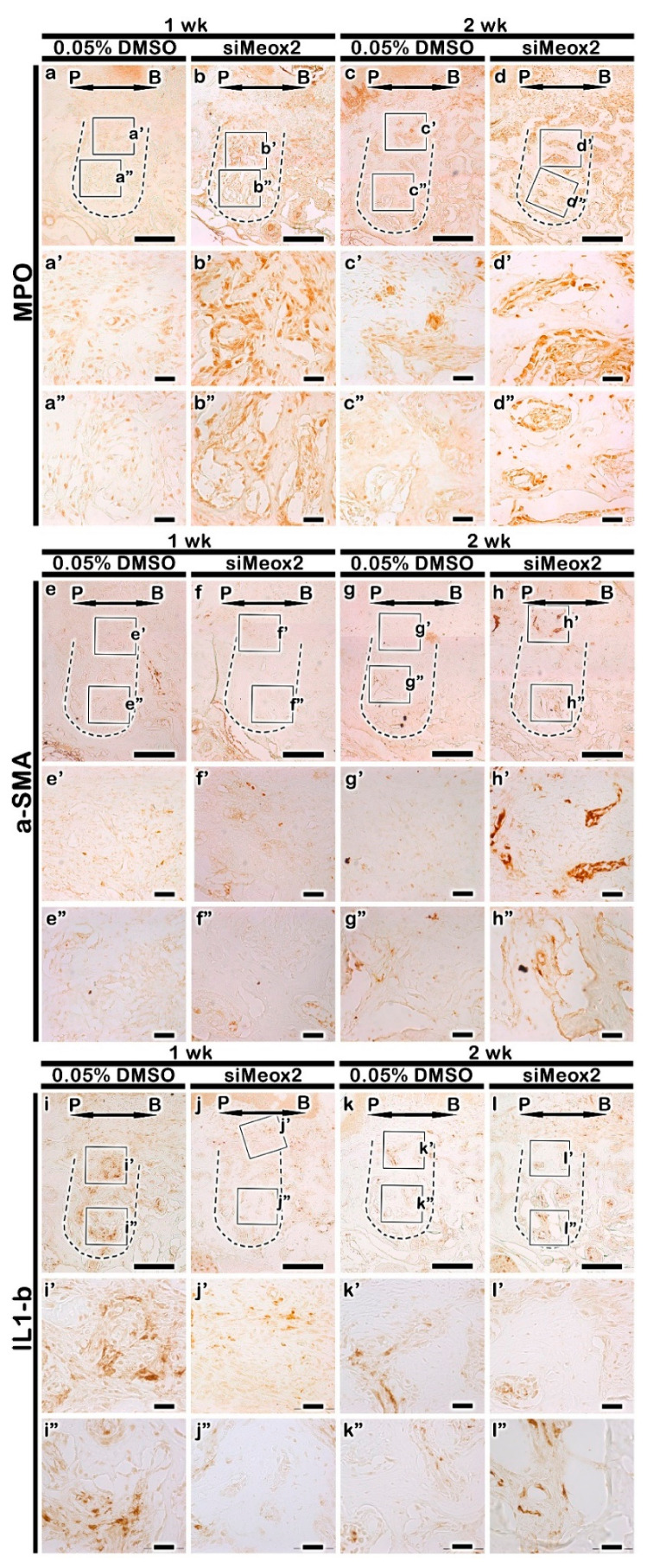
Immunolocalization of Myeloperoxidase (MPO) (**a**–**d**), alpha- Smooth muscle actin (α-SMA) (**e**–**h**) and Interleukin 1 beta (Il-1β) (**i**–**l**). The number of MPO positive cells is higher in si*Meox2* treated specimens (**a**) than in carrier control (**b**) at 1 week. After 2 weeks of treatment, the number of MPO positive cells is lower in carrier control-treated specimens (**c**) than in si*Meox2* treated specimens (**d**). After 1 week of treatment, carrier control (DMSO)-treated specimens display fewer α-SMA positive cells (**e**), while the number of α-SMA positive cells is decrease in si*Meox2* treated specimens (**f**). After 2 weeks of treatment, normal distribution of localization of α-SMA in carrier control-treated specimens (**g**) and si*Meox2* treated specimens (**h**). The number of IL-1β positive cells is higher in carrier control-treated specimens (**i**) than in si*Meox2* treated specimens (**j**). The normal distribution of localization of inflammation in carrier control-treated specimens (**k**) and si*Meox2* treated specimens (**l**) after 2 weeks of treatment. P, palatal and B, buccal. Dotted line demarcates the root region of the socket. The solid boxes depict higher magnification views a′–l″. Scale bars 200 µm (**a**–**l**), 100 µm (**a**′–**l**″).

**Figure 6 ijms-21-08701-f006:**
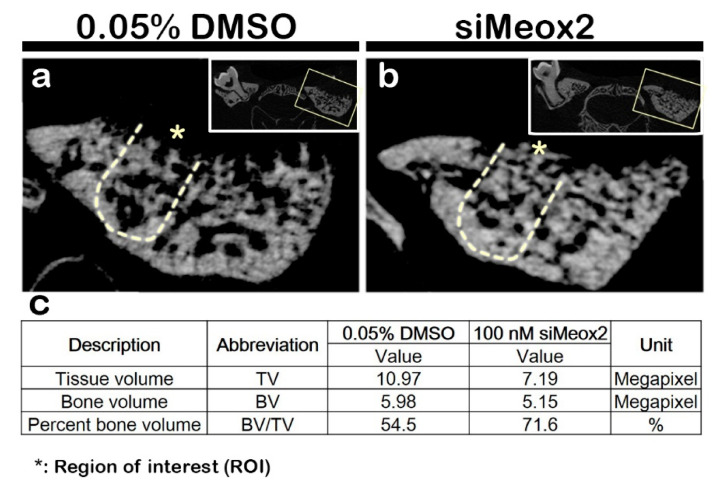
Micro-CT analysis of the tooth root socket and newly formed bone. After 2 weeks, the carrier control (DMSO) treated specimens show less volume of bone formed (**a**) compared with that in the knocking down of *Meox2* (**b**). The solid boxes depict higher magnification views. The table of analysis determined using CTAn software (**c**).

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
