# Peer review of "Facilitation of Bone Healing Processes Based on the Developmental Function of Meox2 in Tooth Loss Lesion"

_ijms, 2020, doi:10.3390/ijms21228701_

Round 1

Reviewer 1 Report

A very interesting and well conducted study. Several aspects appreciated:

_ The rigorous methodology

_ The precise description of the results

_ Recent references

Slight improvements could be made:

_ Enlarge the images in figures 4 and 5 (they are too small).

_ Write a paragraph "conclusions" that summarizes in a general way the scope of the results obtained.

The study does not need further elements to modify.

Author Response

Reviewer 1

Comments and Suggestions for Authors

A very interesting and well conducted study. Several aspects appreciated:

- The rigorous methodology

-The precise description of the results

- Recent references

: Thank you for your consideration.

Slight improvements could be made:

- Enlarge the images in figures 4 and 5 (they are too small).

: As reviewer suggested, we rearranged images in figure 4 and 5.

Page 6, 8

- Write a paragraph "conclusions" that summarizes in a general way the scope of the results obtained.

: As reviewer suggested, we have prepared the conclusion paragraph at the end of the article.

Page 11, Line: 335; Page 12, Line: 337~341

Conclusion

Based on our results, we suggest that the knocking down of Meox2 via the activation of osteoblast and modulation of inflammation would be a plausible answer for bone regeneration as a gene therapy. This study provides important principle dependent application of developmental mechanisms into tissue regeneration would be an excellent model system and gene therapy development for overcoming unsolved diseases.

Reviewer 2 Report

In the article “Facilitation of bone healing processes based on the developmental function of Meox2 in tooth loos lesion” the authors explored the bone healing capacity of Meox2 gene. The authors performed in vitro assays in which they knocked down Meox2 on human periodontal ligament fibroblasts to examine the osteogenic potential of meox2 and in vivo experiment in which authors explored the potential application of Meox2 knockdown in bone healing processes using periodontitis model in mice.

The paper is well written, all experiments are described in detail and results are clearly presented. However, I have few suggestion which might improve the paper:

A) Fig.5 - The figure is composed from a total of 36 subfigures, and the subfigures are small and it is difficult to see the details on them; I would suggest to rearrange this figure or reduce the number of subfigures.

B) Fig.6c - It is unnecessary to show values with 5 decimal places; in this case one decimal place is more than enough.

C) In Material and Methods section the authors should provide the product details uniformly. The authors use several styles of product details: (manufacturer), (manufacturer, country), (manufacturer, city, country); For the names of US federal states the authors use abbreviations but for Nevada they use the full name and also provide postal code of the city.

D) Please describe better the experimental design of the in vivo experiment (section 2.1. Animals). In the sentence “Three mice from each of the remaining groups were randomly selected and sacrificed on the first, second and third weeks of treatment.” It is unclear what are “the remaining groups”. It would be easier to understand the experimental design if it was additionally presented graphically or in table.

Author Response

Reviewer 2

Comments and Suggestions for Authors

In the article “Facilitation of bone healing processes based on the developmental function of Meox2 in tooth loos lesion” the authors explored the bone healing capacity of Meox2 gene. The authors performed in vitro assays in which they knocked down Meox2 on human periodontal ligament fibroblasts to examine the osteogenic potential of meox2 and in vivo experiment in which authors explored the potential application of Meox2 knockdown in bone healing processes using periodontitis model in mice.

The paper is well written, all experiments are described in detail and results are clearly presented. However, I have few suggestion which might improve the paper:

  1. A) Fig.5 - The figure is composed from a total of 36 subfigures, and the subfigures are small and it is difficult to see the details on them; I would suggest to rearrange this figure or reduce the number of subfigures.

: As reviewer suggested, we have rearranged the Figure 5 for better understanding.

Page: 8

  1. B) Fig.6c - It is unnecessary to show values with 5 decimal places; in this case one decimal place is more than enough.

: As reviewer suggested, we have changed the decimal place and unit of pixel in the figure 6 for better understanding.

Page: 9

  1. C) In Material and Methods section the authors should provide the product details uniformly. The authors use several styles of product details: (manufacturer), (manufacturer, country), (manufacturer, city, country); For the names of US federal states the authors use abbreviations but for Nevada they use the full name and also provide postal code of the city.

: We rewrote the product details: (manufacturer, country) in the manuscript, as reviewer suggested.

Page 12, Line: 349~379; Page 13, Line: 391~437, Page 14, Line: 438~453

  1. D) Please describe better the experimental design of the in vivo experiment (section 2.1. Animals). In the sentence “Three mice from each of the remaining groups were randomly selected and sacrificed on the first, second and third weeks of treatment.” It is unclear what “the remaining groups” are. It would be easier to understand the experimental design if it was additionally presented graphically or in table.

: As reviewer suggested, we have prepared the table for better understanding of experimental design. We have included the table as a supplementary file (Supplementary Table 2). In addition, we rewrote the sentences in the materials and method section for better understanding.

Page 12, Line 382~385; Page 13, Line: 386~387

Adult 8 week-old male B6 mice were used in the experiment (n = 24). The mice were assigned as the control group (n = 4 per week, a total of 12 for 3 weeks) or the experimental group (n = 4 per week, a total of 12 for 3 weeks) with treatment following ligature-induced periodontitis and tooth extraction under anesthesia. Three mice from each of groups were randomly selected and examined the micro-CT evaluation on the second week first, then processed for further evaluation including histology and immunohistochemistry.
